# Independence of Peripheral Brain-Derived Neurotrophic Factor from Depression and Anxiety Symptoms in Cocaine Use Disorder: An Initial Description

**DOI:** 10.3390/ijms26178294

**Published:** 2025-08-27

**Authors:** Dannia M. Islas-Preciado, Ruth Alcalá-Lozano, Erika P. Aguilar-Velazquez, Yvonne G. Flores-Medina, Nelly M. Vega-Rivera, Erika Estrada-Camarena, Ruben Carino-Escobar, Jorge J. González-Olvera, Erik D. Morelos-Santana

**Affiliations:** 1Laboratorio de Neuromodulación, Subdirección de Investigaciones Clínicas, Instituto Nacional de Psiquiatría Ramón de la Fuente Muñiz, Ciudad de México 14370, Mexico; dislas@inprf.gob.mx (D.M.I.-P.); ruthalcala@inprf.gob.mx (R.A.-L.); paola.1428av@gmail.com (E.P.A.-V.); 2Facultad de Medicina, Universidad Nacional Autónoma de México, Ciudad de México 04510, Mexico; 3Subdirección de Investigaciones Clínicas, Instituto Nacional de Psiquiatría Ramón de la Fuente Muñiz, Ciudad de México 14370, Mexico; yg.floresmedina@inprf.gob.mx (Y.G.F.-M.); jjgonz@gmail.com (J.J.G.-O.); 4Laboratorio de Neuropsicofarmacología, Dirección de Investigaciones en Neurociencias, Instituto Nacional de Psiquiatría Ramón de la Fuente Muñiz, Ciudad de México 14370, Mexico; nmvega@inprf.gob.mx (N.M.V.-R.); estrada@inprf.gob.mx (E.E.-C.); 5División de Investigación en Neurociencias Clínicas, Instituto de Rehabilitación Luis Guillermo Ibarra Ibarra, Ciudad de México 14839, Mexico; rubencarinoe@hotmail.com; 6Facultad de Psicología, Universidad Nacional Autónoma de México, Ciudad de México 04510, Mexico

**Keywords:** depression, anxiety, BDNF, cocaine

## Abstract

Cocaine use disorder (CUD) presents high comorbidity with mood symptoms that impair recovery processes and facilitate relapses. Peripheral brain-derived neurotrophic factor (BDNF) is negatively correlated with mood symptoms in depressive disorders. However, whether a correlation exists between BDNF and mood in CUD is still unknown. Thus, in this cross-sectional study, we explored the potential relationship between peripheral BDNF levels and depression and anxiety symptoms in CUD. Serum peripheral BDNF was determined by the ELISA method. Standardized Hamilton Depression (HDRS) and Anxiety (HARS) inventories were administered. Twenty-nine seeking-treatment CUD participants under stable medication (female = 3) were enrolled. According to the mood severity, 34.48% of participants were classified as normal, 24.14% as moderate, and 41.38% as severe symptoms (*p* < 0.001). Peripheral BDNF was similar between the different mood severity groups (*p* > 0.05). No correlation between BDNF and HDRS and BDNF and HARS was detected regardless of the severity of mood symptoms (*p* > 0.05). Different from what has been observed in depressive disorders, independence between peripheral BDNF levels and mood symptoms in CUD was observed. This finding suggests a singular, intricate regulation of peripheral BDNF and mood as part of CUD-related maladaptations that might disrupt the expected antidepressant response and perpetuate mood symptoms in CUD.

## 1. Introduction

Cocaine and crack consumption is a major public health concern worldwide, with a prevalence estimated at 0.42% of the general population [1]. Of concern, cocaine consumption is increasing as global production has raised by 20% compared to 2021 [2]. Chronic cocaine/crack consumption may lead to cocaine use disorder (CUD), characterized by a cyclic pattern of consumption, loss of control, withdrawal, and relapse [3,4] that conforms with an addictive cycle. Previous research has shown high comorbidity between depressive symptoms and substance consumption, such as alcohol, marijuana, and cocaine [5]. Also, during different stages of the addictive cycle, increased comorbidity between depressive and anxiety symptoms presents in CUD [6,7], interfering with the recovery process and facilitating relapses [8]. Worryingly, up to 80% of people with CUD have reported experiences of affective symptoms such as major depression, which increases the risk of relapse [9]. Preclinical studies have similarly shown increased anxiety-like behaviors during the cocaine withdrawal phase [10].

Mood is strongly modulated by neurotrophins, particularly brain-derived neurotrophic factor (BDNF) [11]. BDNF is a neurotrophin synthesized and distributed along the central and peripheral systems that is involved in neuronal structure and plasticity [12]. Also, BDNF is strongly implicated in brain structures, such as the hippocampus, cerebral cortex, and amygdala, that are critical for mood regulation [13,14]. Previous clinical evidence has reported an inverse relation between peripheral BDNF and mood symptoms [15,16,17]: when BDNF increases, lower depressive scores in the Beck Depressive Inventory (BDI) and Hamilton Rating Scale for Depression (HDRS) can be observed. The link between BDNF and depression is strong, as postmortem reports have found decreased *BDNF* gene expression in the brains of people with depression symptoms compared with those of non-depressed individuals [18]. In addition, the administration of antidepressant agents, such as selective serotonin reuptake inhibitors (SSRIs) or serotonin and norepinephrine reuptake inhibitors (SNRIs), increases BDNF peripheral levels, which correlate with improvement in depressive symptoms [19,20]. Overall, the literature suggests a significant relationship between BDNF peripheral levels and mood symptoms.

In contrast, BDNF seems to play a more complex role in CUD. For example, increases in BDNF activity in the rat hippocampus, measured as an elevation of mRNA and BDNF protein during cocaine withdrawal, parallel depressive-like behaviors [21]. In humans, some studies have revealed higher peripheral BDNF levels in cocaine users compared to control subjects [22] and higher blood BDNF in subjects with CUD-induced depressive disorder [23]. However, it is not known whether there is a relationship between BDNF and mood symptoms in CUD. Due to this, the present study aimed to examine the relationship between BDNF peripheral levels and depression and anxiety symptoms in people with CUD. In addition, we aimed to compare BDNF peripheral levels between different severity categories of depression and anxiety symptoms in persons with CUD.

## 2. Results

Twenty-nine participants were included (female n = 3). Based on HDRS scores, subjects were grouped into normal, moderate, and severe depression categories. Sociodemographic characteristics are presented in Table 1. No significant differences in age, years of education, employment, or marital status were detected among the groups. Similarly, no significant differences were found for urine tests for crack or cocaine consumption, nor were any differences observed in alcohol, tobacco, cannabis, or methamphetamine usage.

### 2.1. Addiction Severity Index-6 (ASI-6)

ASI-6 scores ranging between 0 and 1 showed no significant differences in medical, legal, employment, alcohol, or drug domains between severity levels registered in the HDRS (Table 1). This suggests similar levels of severity of addiction among the groups. However, the domain of familiar issues showed significant differences between the mood severity groups, where the severe group had the highest score in this domain. The results are detailed in Table 1.

### 2.2. Depressive and Anxiety Symptoms

HDRS showed a score of 5 points (±1.15) for the normal category; for moderate severity, a mean of 13.14 (±2.5) points was scored, and for the severe group, a mean of 25.17 (5.9) points was registered, and significant differences between levels of symptom severity were observed (F_2,26_ = 66.1; *p* < 0.001; η^2^ = 0.84). Regarding the Hamilton Anxiety Rating Scale (HARS), a mean of 6.9 (4.0) was registered for the normal category; for the moderate group, a mean of 23.7 (9.2) points was registered, and for the severe group, a mean of 31.7 (12.1) points was scored. Similarly to the HDRS results, HARS showed significant differences between the levels of severity reported (F_2,26_ = 19.5; *p* < 0.001; η^2^ = 0.60).

### 2.3. Peripheral BDNF Levels and Depressive and Anxiety Symptoms

The mean of peripheral serum BDNF for the normal group was 32.8 ng/mL (±12.6); for the moderate severity group, it was 32.7 ng/mL (9.49), and for the severe group, it was 30.02 ng/mL (9.54). Non-parametric ANOVA showed no differences between the normal, moderate, and severe symptoms groups (F_2,26_ = 0.12, *p* = 0.8, η^2^ < 0.05). Additionally, correlations between serum BDNF and depressive symptoms showed no significant relationship between the normal (r = 0.24, t = 0.71, df = 8, *p* = 0.4), moderate (ρs = 0.17, *p* = 0.7), and severe groups (r = 0.4, t = 1.38, df = 10, and *p* = 0.19) (Figure 1). Moreover, correlations between anxiety symptoms registered in HARS and BDNF levels were not statistically significant among the normal (r = 0.34, t = 1.0, df = 8, *p* = 0.3), moderate (ρs = 0.17, *p* = 0.7), and severe groups (r = 0.4, t = 1.3, df = 10, *p* = 0.1).

In a further categorization, a comparison between normal (non-depressed) and depressed participants (moderate and severe) was made. Similarly, BDNF peripheral levels did not differ between those two groups (t = 0.41, df = 14.2, and *p* = 0.6). In addition, BDNF levels and depression symptoms showed no significant correlations between the normal (r = 0.24, t = 0.71, df = 8, and *p* = 0.4) and depressed groups (r = 0.01, t = 0.07, df = 17, and *p* = 0.9), and HARS correlations showed similar results.

Correlations between peripheral BDNF levels and HARS severity categorization showed no significant association between the normal (ρs was unable to be computed due to a low sample), mild (r < 0.01, t < 0.01, DF = 8, and *p* = 0.9), and severe groups (r = −0.15, t = −0.4, DF = 9, and *p* = 0.6) (Figure 2).

### 2.4. Substance Consumption

A higher number of participants had positive recent cocaine consumption in the moderate and severe groups (Table 1), but the opposite was observed in the normal non-depressed subjects. However, these differences were not statistically significant. The results are included in Table 1.

Regarding consumption of other substances, such as alcohol, tobacco, cannabis, or methamphetamine, no significant differences were found between the normal, moderate, and severe depression symptom groups.

## 3. Discussion

In this study, peripheral BDNF levels and depressive or anxiety symptoms in CUD were not correlated. This independence is regardless of the severity of symptoms, and no difference in peripheral BDNF levels was found between normal, moderate, and severe symptoms of anxiety and depression.

In contrast to the observed in depressive disorders, where a negative correlation has been detected between BDNF and the scores of depressive symptoms [15,19,24,25], this was not observed in our sample of participants with CUD, who scored moderate or severe depressive symptoms. Although BDNF is strongly involved in mood regulation, this cross-talk may not be straightforward in CUD.

Robust evidence shows a negative correlation between depression scores and peripheral BDNF levels in unmedicated MDD [15], and more recent meta-analyses have confirmed those findings [24,25]. The neurotrophic hypothesis of depression proposes that BDNF depletion contributes to mood dysregulation, and pharmacotherapy restores BDNF levels [26] to those observed in non-depressed people [27]. Also, other rapid-acting antidepressant treatments have shown similar BDNF elevations [28], supporting the notion that upregulating BDNF along with treatments improves mood symptoms. However, the uncorrelation observed here between depressive symptoms and peripheral BDNF suggests that mood symptoms in CUD might not be fully aligned with the neurotrophic hypothesis. In support of this notion, comorbid cocaine MDD-induced or MDD-primary CUD individuals showed similar peripheral BDNF levels in a depression remission period, showing no difference in BDNF regardless of MDD chronicity in CUD [23].

While antidepressants typically elevate BDNF levels [29], their effect appears to be limited in depression and anxiety symptoms in CUD [30,31]. In our CUD sample, 65% of the participants were under a stable pharmacological dose of antidepressants (>one month); however, they continued experiencing moderate to severe depression and anxiety symptoms. This may indicate a partial antidepressant response in our participants and may help to explain the independent BDNF fluctuation with respect to mood in CUD. Moreover, Pettorruso et al. [22] also showed that psychopharmacological treatment is associated with higher levels of BDNF. A previous meta-analysis suggests that after antidepressant treatment, the negative correlation between pre-treated depression and BDNF disappears [15], which could help to partially explain the uncorrelation described here, but not clarify the similar levels of BDNF among treated normal, moderate, and severe depression CUD symptom groups.

The peripheral BDNF levels apparently dissociated from mood symptoms observed here may also be involved with dopaminergic activity in the reward system. Dopaminergic activation heightens BDNF trafficking and gene expression [32]. Animal models have shown that increases in BDNF in key structures of the reward system, such as the ventral tegmental area (VTA) and the nucleus accumbens, elicit cocaine-seeking behaviors [33,34]. Those BDNF elevations underlie the maintenance of seeking behaviors through learning and memory consolidation mechanisms [35], which is crucial to preserving the addictive cycle. A previous report shows peripheral BDNF upregulation in chronic cocaine use [22] in CUD treatment-seeking subjects compared to healthy controls, and elevations in this factor have predicted relapses [36,37]. Thus, elevations in BDNF could be part of a mechanism of maladaptive responses underlying relapses in CUD. Supporting this notion, elevated BDNF positively correlates with craving [38], which, in turn, facilitates relapses. Overall, those findings add evidence of the potential negative role of BDNF increases at key phases of the CUD recovery process.

From preclinical evidence, it is known that recent cocaine consumption increases BDNF in the reward system [39,40]. However, we found no difference in urine-positive tests among groups, suggesting that recent consumption may not influence our peripheral BDNF results. Despite methodological differences when evaluating BDNF, previous studies have shown a positive correlation between peripheral and central BDNF levels [41]. Thus, it is feasible to suggest that recent cocaine consumption may not influence our peripheral BDNF results, but region-specific fluctuations should not be fully disregarded.

Our findings could add evidence relevant to CUD management by revealing that depression and anxiety symptoms could persist despite pharmacological treatment without influence on BDNF peripheral levels. Thus, by not observing BDNF expectable patterns, perhaps due to possible disruptions associated with cocaine use adaptations, such as repeated activation of the reward system, the antidepressant treatment response may differ in CUD. Also, those adaptations in the reward system could further explain the independence from peripheral BDNF and mood symptoms observed here. Novel interventions that consider the potential unique pattern of BDNF in CUD are necessary to counteract mood symptoms caused by cocaine consumption. Notably, high depression and anxiety symptoms that persist in CUD may facilitate relapses and impair recovery [6,8].

BDNF protein activates tropomyosin receptor kinase B (TrkB), and acute and sustained cocaine consumption upregulates this receptor in the reward system [42,43]. Some interventions, such as repetitive transcranial magnetic stimulation (rTMS), elevate peripheral BDNF in MDD after an antidepressant response [44,45]. Regarding the TrkB receptor, the opposite effect has been described in people with MDD and suicidal ideation [45]. Also, Ref. [44] found no correlation in HDRS or Montgomery-Åsberg Depression Rating Scale (MADRS) scores and the TrkB receptor, suggesting a potential non-relationship between depression symptoms and the TrkB receptor in MDD.

On the other hand, the pro-BDNF/BDNF ratio has been suggested as indicative of an imbalance in plasticity and neurogenerative activity in MDD [46]. The work of Miuli et al. [47] found a negative correlation between the pro-BDNF/BDNF ratio and insomnia in CUD, but similar to our work, they reported no association between the pro-BDNF/BDNF ratio and MADRS depression scores. Finally, Pettorruso et al. [22] described an equivalent pro-BDNF/BDNF ratio between CUD participants and HC controls, along with the mentioned BDNF elevations. Thus, although the pro-BDNF/BDNF ratio might indicate a neurotrophin imbalance in some conditions, this interpretation could be limited in CUD due to the potential upregulation of the BDNF system previously discussed.

In this study, we found that the familiar problems domain from ASI-6 had a significantly higher index in the severe group with respect to the moderate and normal depression groups. Given that a familiar environment represents the individual’s primary and most immediate social context, we might expect to observe significant differences in this domain. Specifically, it is suggested that elevated depressive symptomatology may disrupt family dynamics, leading to challenging interactions between its members, which could be exacerbated by substance use [48].

## 4. Materials and Methods

### 4.1. Study Design

The data for this report are baseline measurements of participants with CUD enrolled in a randomized clinical trial that aims to determine the effects of a non-invasive brain stimulation technique on BDNF- and CUD-related outcomes (Clinicaltrials.gov NCT06189690). Thus, a cross-sectional design for this report is considered. The study was conducted in accordance with the guidelines of the Declaration of Helsinki and was approved by local ethics and research committees (CEI/C/028/2019; IC16036.2).

### 4.2. Participants

Ambulatory seeking-treatment participants were recruited from the Addictions Clinic of the National Institute of Psychiatry Ramon de la Fuente Muñiz and from outpatient public services. Participants gave written informed consent before enrollment. Inclusion criteria were as follows: subjects from 18 to 50 years old, at least 6 years of formal education (equivalent to elementary school), and a diagnosis of cocaine or crack use disorder according to DSM-5 [3]. The exclusion criteria comprised diagnosis of neurological illness; major mental disorders like schizophrenia, bipolar disorder, or obsessive–compulsive disorder; or any substance use disorder except alcohol or tobacco use disorder. Elimination criteria included the explicit desire to withdraw from the study. Given that pharmacological agents can modify BDNF levels [15], at least one month of stable medication before enrollment was required. Also, interruptions of crack/cocaine consumption may modify BDNF levels [49,50]; therefore, a cocaine-urine test was performed to inquire about recent consumption. We considered a non-consumer participant to be an individual with a negative result in the cocaine-urine test on the day when the blood was sampled.

### 4.3. Clinical Assessments

A qualified psychiatrist conducted all diagnostic and clinical evaluations. Inclusion and exclusion criteria were confirmed using the Mini-International Neuropsychiatric Interview, MINI Plus [51]. Severity was assessed by the Addiction Severity Index (ASI-6) [52], which assigns scores to problems in familiar/social, work, legal, medical, and psychosocial contexts, as well as alcohol and drug-related issues. Higher indexes reflect a higher severity rating, from 0 to 1.

Depressive state was assessed using the 17-item Hamilton Depression Rating Scale (HDRS), which considers normal (<8 points), moderate (<18 points), and severe (>18 points) depression based on a score obtained [53]. Anxiety was assessed with the 14-item Hamilton Anxiety Rating Scale (HARS), which considers normal (<13), moderate (<24 points), and severe anxiety (>25 points) based on the total score obtained [54].

### 4.4. Blood Sampling and Serum Preparation

Blood was collected from the medial cubital vein at 8:00 a.m. following 8 h fasting, using 10 mL anticoagulant-free tubes (Becton Dickinson, NY, USA). After collection, samples were clotted for 1 h at room temperature and then centrifuged at 2500 RPM for 10 min to extract serum. Serum samples were aliquoted in 0.5 mL tubes and stored at −80 °C until BDNF determination.

After blood collection, a sample of urine was collected in a plastic container, where the sample was processed with a multi-drug assay (Alpha Laboratories, Poway, CA, USA). The assay determined the recent consumption of cocaine and other substances, such as opioids and methamphetamines.

### 4.5. Peripheral BDNF Quantification

BDNF serum levels were determined with an enzyme-linked immunosorbent assay (ELISA) commercial kit (ID product CYT306, Sigma-Aldrich, St. Louis, MO, USA) in accordance with the manufacturer’s instructions. The standard curve and all samples were assayed in duplicate. Intra- and inter-assay variations were ±3.7% and ±8.5%, respectively, with a range of detection between 15 and 1000 ng/mL.

### 4.6. Data Analysis

Data were analyzed with R version 3.6.1 [55] and R Studio version 2024.12.1 [56]; plotting was performed with the ggplot2 package version 3.5.1 [57]. One-way ANOVA was performed to analyze HDRS and HARS scores. The Aligned Ranged Test Tool (ARTool) version 0.11.1 [58] was used to perform non-parametric ANOVA, determining the level of HDRS severity (normal, moderate, or severe). Categorical variables were statistically analyzed with the Chi-squared (X2) test. The relationship between mood symptoms and BDNF levels was assessed through Pearson’s (r) or Spearman’s correlation (ρs) when applicable. Normality of the data was evaluated using the Shapiro–Wilk test. In instances where the assumption of normality was not met, a non-parametric alternative test was applied. Means and standard deviation (±) are presented as numerical variables. No pre-specified sample size calculations were performed in this cross-sectional report, and the sample size was limited to participants enrolled in the randomized clinical trial. However, a post hoc power analysis was performed in G*Power 3.1 [59] to assess the risk of type II error for the observed results. The statistical power reached was 99.3% for the correlations between total BDNF and HDRS scores using the following parameters: test family = exact, test = correlation (bivariate normal model), two tails, α = 0.05, n = 19 (moderate and severe depression symptoms groups), and H0 = 0.8.

## 5. Conclusions

We found no correlation between BDNF peripheral levels and mood symptoms in CUD, apparently differing from mood disorders. This finding may not be aligned with the neurotrophic hypothesis of depressive symptoms. The independence of BDNF levels from depression and anxiety symptoms that occurred across three different levels of mood severity suggested that chronic cocaine consumption might intricate a relationship between BDNF and mood regulation in CUD. Consequently, the expected antidepressant response and typical BDNF increases could be disrupted in CUD, and this may limit the effectiveness of pharmacological approaches, perpetuating mood symptoms in CUD. Thus, it is crucial to consider BDNF activity to better understand some of the underlying processes that lead to timely, effective interventions to improve mood in CUD.

### Limitations

The main limitation of the present study regards the sample included, as it was composed mainly of male subjects. The subjects included here were voluntarily seeking treatment for CUD at a local institution where, consistent with the previous literature, substance-use-disorder-treatment-seeking populations are predominantly male [60,61]. Several factors may explain the gender imbalance in people seeking CUD treatment. This may be related to the fact that women develop CUD later than men [62], and they are more frequently diagnosed with other mental disorders [63]. Moreover, social stigmatization and marginalization of women relating to drug abuse limit access to recovery services [60,64,65]. Additionally, socioeconomic differences, such as lower income among women [66], make it difficult for them to access treatment. A similar proportion of women and men is reported in previous studies on CUD [22,62], even with a larger sample size (n > 1000) [67]. Thus, the sample included here could be considered representative of what has been observed in the CUD seeking-treatment population, but it underrepresents females, limiting the generalizability of our results.

## Figures and Tables

**Figure 1 ijms-26-08294-f001:**
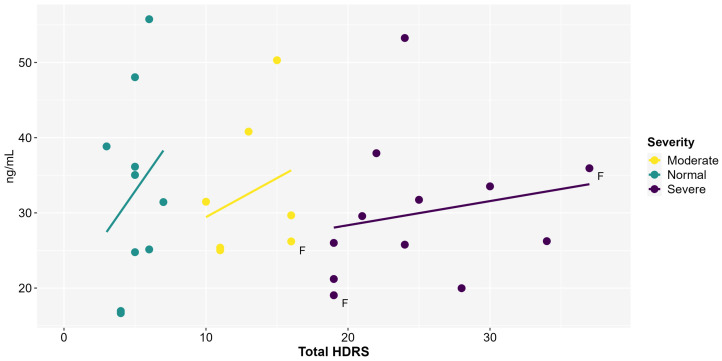
Peripheral BDNF concentrations (ng/mL) and total Hamilton Depression Rating Scale (HDRS) score (n = 29). Each line represents the best-fit adjustment between variables separately for the three different levels of depression symptom severity. Pearson’s correlation was not significant. F = female.

**Figure 2 ijms-26-08294-f002:**
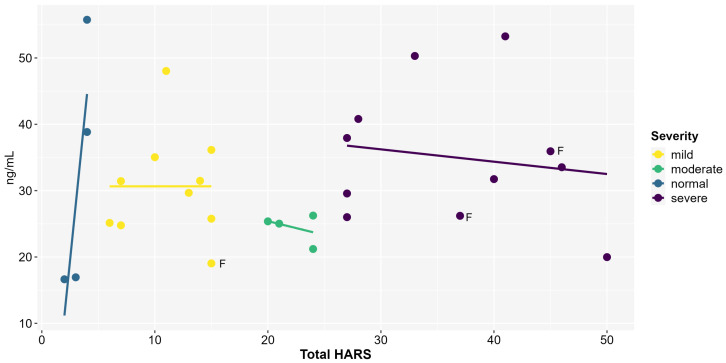
Peripheral BDNF concentrations (ng/mL) and total Hamilton Anxiety Rating Scale (HARS) score (n = 29). Each line represents the best-fit adjustment between variables separately for the four levels of anxiety symptom severity. Pearson’s correlation was not significant. F = female.

**Table 1 ijms-26-08294-t001:** Sociodemographic, treatment, and severity of addiction among mood severity groups.

	Normal (n = 10) HDRS 3–7 Points	Moderate (n = 7) HDRS 10–16 Points	Severe (n = 12) HDRS 19–37 Points	F/X^2^, *p*
**Sociodemographic**				
Age	31.1 (6.8)	32.5 (7.8)	32.7 (8.8)	F_2,26_ = 0.103, *p* = 0.9
Education	12.6 (2.8)	15.0 (4.4)	14.0 (3.0)	F_2,26_ = 1.09, *p*= 0.35
Employment (n)	Full-time (3)	Full-time (1)	Full-time (5)	X^2^ = 4.897, df = 3, *p* = 0.5
	Freelance (5)	Freelance (2)	Freelance (2)	
	Unemployed (1)	Unemployed (1)	Unemployed (4)	
	Other (1)	Other (3)	Other (1)	
Marital Status (n)	Single (5)	Single (5)	Single (7)	X^2^ = 1.47, df = 2, *p* = 0.83
	Married (2)	Married (1)	Married (1)	
	Other (3)	Other (1)	Other (4)	
**Treatment**				
Psychological (n)	(5)	(3)	(5)	X^2^ = 0.16, df = 2, *p* = 0.9
Pharmacological (n)	(9)	(5)	(10)	X^2^ = 2.9, df = 2, *p* = 0.5
SSRIs	(6)	(4)	(8)	X^2^ = 0.19, df = 2, *p* = 0.3
SNRIs	(2)	(1)	(0)	NC
Anticonvulsants	(10)	(4)	(9)	X^2^ = 3, df = 4, *p* = 0.5
Antipsychotics	(6)	(4)	(6)	X^2^ = 0.3, df = 4, *p* = 0.3
**ASI-6 domains**				
Medical	0.1 (0.18)	0.23 (0.36)	0.14 (0.26)	F_2,26_ = 0.5, *p* = 0.5
Legal	0.02 (0.06)	0.07 (0.13)	0.03 (0.12)	F_2,26_ = 0.7, *p* = 0.4
Employment	0.62 (0.24)	0.5 (0.33)	0.69 (0.26)	F_2,26_ = 0.7, *p* = 0.4
* Familiar	0.26 (0.13)	0.22 (0.08)	0.32 (0.18)	F_2,26_ = 1.02, *p* = 0.03
Alcohol	0.18 (0.11)	0.15 (0.16)	0.15 (0.17)	F_2,26_ = 0.08, *p* = 0.9
Drugs	0.19 (0.1)	0.18 (0.06)	0.2 (0.06)	F_2,26_ = 0.1, *p* = 0.8
**Crack/Cocaine consumption**				
Status cocaine urine test (n)	+ (4) − (6)	+ (6) − (1)	+ (9) − (3)	X^2^ = 4.6, df = 2, *p* = 0.09
Years of consumption	10.8 (5.6)	6.6 (6.7)	6.6 (5.2)	F_2,26_ = 1.66, *p* = 0.2
**Other substances**				
Alcohol (n)	(2)	(2)	(3)	X^2^ = 0.17, df = 2, *p* = 0.9
Tobacco (n)	(5)	(5)	(9)	X^2^ = 1.6, df = 2, *p* = 0.4
Cannabis (n)	(0)	(2)	(3)	X^2^ = 3.2, df = 2, *p* = 0.2
Methamphetamine (n)	(0)	(1)	(0)	X^2^ = 3.2, df = 2, *p* = 0.19

HDRS: Hamilton Depression Rating Scale. ASI-6: Addiction Severity Index-6. SSRIs: selective serotonin reuptake inhibitors. SNRIs: serotonin and norepinephrine reuptake inhibitors. Standard deviations are presented inside ( ) or as n when it is specified. * Statistically significant. NC: Non-computable. + indicates cocaine-positive urine test. − indicates cocaine-negative urine test.

## Data Availability

The raw data supporting the conclusions of this article will be made available by the authors upon reasonable request.

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
