# Peer review of "Independence of Peripheral Brain-Derived Neurotrophic Factor from Depression and Anxiety Symptoms in Cocaine Use Disorder: An Initial Description"

_ijms, 2025, doi:10.3390/ijms26178294_

Round 1
Reviewer 1 Report
Comments and Suggestions for Authors
The manuscript entitled „Independence of peripheral brain derived neurotrophic factor from depression and anxiety symptoms in cocaine use disorder“ by Islas-Preciado and coworkers estimates the relationship between peripheral BDNF and mood disorders on the basis of cocaine use disorder. The basic idea is good, but I have some serious concerns.
The sample size is low.
To rely on peripheral BDNF levels as a single biomarker is very questionable.
BDNF is just one of the neurotrophin system elements:
- The absolute levels of BDNF should be analyzed along with its precursor (proBDNF) since they have the opposite actions, and therefore their ratio is a qualitatively more important indicator (this could be analyzed in peripheral blood samples)
- the beneficial effects of BDNF are achieved via TrkB receptors, so their expression is equally important as BDNF levels (this could also be analyzed in peripheral blood samples)
This manuscript, at this stage, does not meet the criteria for publication. However, with the suggested additional analyses, it can be substantially improved and resubmitted.
Reviewer 2 Report
Comments and Suggestions for Authors
Comments to the Authors
Manuscript ID: ijms-3659027
Title: Independence of peripheral brain derived neurotrophic factor from depression and anxiety symptoms in cocaine use disorder.
The Authors aimed to examine the relationship between BDNF peripheral levels and depression and anxiety symptoms in people with CUD. In addition, they aimed to compare BDNF peripheral levels among different severity categories of depression and anxiety symptoms in persons with CUD.
Previous knowledge indicates gender differences in BDNF levels, as well as a relationship between gender and mood. It is also known that sex hormones modulate BDNF expression and function.
According to recent literature, there are clear gender differences in cocaine addiction, and consequently differences in the effects of cocaine use. Twenty-nine people participated in the study, including three women. The study did not take into account gender division, and to obtain reliable results, first of all, a larger group of volunteers is needed.
In my opinion, the manuscript is not ready for printing. In order to obtain reliable results, the authors must increase the group of volunteers, taking into account the division by gender.
Round 2
Reviewer 1 Report
Comments and Suggestions for Authors
Aside from the authors agreeing with my comments, I don’t see any additional breakthrough considering the quality and novelty presented in the manuscript. Thus, I can only recommend the papers published in Molecular Psychiatry (https://doi.org/10.1038/s41380-023-02367-7) and in Frontiers in Psychiatry (https://doi.org/10.3389/fpsyt.2022.836771) as a direction to improve the research (i.e., manuscript) to meet the high-quality standards of the Journal.
Author Response
Reviewer 1
R2
Aside from the authors agreeing with my comments, I don’t see any additional breakthrough considering the quality and novelty presented in the manuscript. Thus, I can only recommend the papers published in Molecular Psychiatry (https://doi.org/10.1038/s41380-023-02367-7) and in Frontiers in Psychiatry (https://doi.org/10.3389/fpsyt.2022.836771) as a direction to improve the research (i.e., manuscript) to meet the high-quality standards of the Journal.
R= We appreciate the reviewer’s suggestions. We are aware of the limitations of our work; however, we consider the present description may serve as an initial point to expand the evidence regarding BDNF dynamics and its role in depressive symptoms in CUD. We broadly discuss the independence between BDNF and mood symptoms, while other studies such as Miuali et al., (2022), reported no correlation in CUD; however, do not fully explain or propose potential underlying explanations on why BDNF is not in line with the neurotrophic hypothesis of depression.
Undoubtedly, important knowledge can be extracted from the recommended papers Petorruso et al. (2024) and Fonseca et al. (2022). Yet our study helps to complement their findings by describing BDNF levels in people with recent cocaine consumption and active depression symptomatology, differing from Fonseca et al., and the correlational objective of BDNF and depression, differing from Petorruso et al. We appreciate the recommendations of the mentioned papers, and they are included in the discussion section to support our findings. Fonseca et al., (2022) was incorporated in lines 166-169.

Round 3
Reviewer 1 Report
Comments and Suggestions for Authors
The authors commented adequately to my remarks.
Author Response
Authors appreciate the reviewer's comments that help to enhance our manuscript.